# The Role of Tumor Metabolic Reprogramming in Tumor Immunity

**DOI:** 10.3390/ijms242417422

**Published:** 2023-12-13

**Authors:** Xianhong Zhang, Weiguo Song, Yue Gao, Yu Zhang, Yuqi Zhao, Shuailin Hao, Ting Ni

**Affiliations:** State Key Laboratory of Reproductive Regulation and Breeding of Grassland Livestock, Institutes of Biomedical Sciences, School of Life Sciences, Inner Mongolia University, Hohhot 010070, China; zhangxianhong96@163.com (X.Z.); song7465361@163.com (W.S.); gyueyue1231@163.com (Y.G.); 13353442149@163.com (Y.Z.); zhao0305495@163.com (Y.Z.)

**Keywords:** tumor metabolism, tumor microenvironment, immune cells, tumor immunity, tumor therapy

## Abstract

The occurrence and development of tumors require the metabolic reprogramming of cancer cells, namely the alteration of flux in an autonomous manner via various metabolic pathways to meet increased bioenergetic and biosynthetic demands. Tumor cells consume large quantities of nutrients and produce related metabolites via their metabolism; this leads to the remodeling of the tumor microenvironment (TME) to better support tumor growth. During TME remodeling, the immune cell metabolism and antitumor immune activity are affected. This further leads to the escape of tumor cells from immune surveillance and therefore to abnormal proliferation. This review summarizes the regulatory functions associated with the abnormal biosynthesis and activity of metabolic signaling molecules during the process of tumor metabolic reprogramming. In addition, we provide a comprehensive description of the competition between immune cells and tumor cells for nutrients in the TME, as well as the metabolites required for tumor metabolism, the metabolic signaling pathways involved, and the functionality of the immune cells. Finally, we summarize current research targeted at the development of tumor immunotherapy. We aim to provide new concepts for future investigations of the mechanisms underlying the metabolic reprogramming of tumors and explore the association of these mechanisms with tumor immunity.

## 1. Introduction

Tumor cells can perform a wide array of metabolic activities via metabolic reprogramming, thus providing energy and a variety of substrates to enable their rapid proliferation and survival [1]. Therefore, abnormal changes in energetic metabolism are important markers of malignant tumors [2]. Tumor metabolic reprogramming not only plays an important role in maintaining tumorigenesis, tumor development, and signal transduction within tumors, but also contributes to regulating the antitumor immune response [3]. Previous studies have found that tumor growth is closely related to the tumor microenvironment (TME). The abnormal accumulation of metabolites produced by the massive proliferation of tumor cells can promote their growth [4]. Furthermore, cancer cells can remodel the TME by secreting various cytokines, chemokines, and other factors to maintain cancer cell proliferation [5]. This implies that the metabolites and cytokines produced by cancer cell metabolism can affect tumor proliferation while also exerting a mutual regulatory function on the TME.

The TME is an important factor affecting the treatment of tumors and features several important components, particularly immune cells. Changes in the metabolic activity of immune cells in the TME, or the destruction of the immune system, can lead to tumor deterioration. Increasing evidence suggests that cancer cells can inhibit the antitumor immune response by competing for and consuming vital nutrients or by reducing the metabolic adaptability of tumor-infiltrating immune cells [6]. In addition, many metabolites in the TME can also affect the differentiation of immune cells and the function of effector factors [7]. More importantly, immune cells can sense various signals in the TME to initiate specific immune functions. Therefore, interfering with the metabolism of tumor cells and immune cells is expected to improve the efficacy of immunotherapy.

## 2. Metabolic Reprogramming of Cancer Cells

The most significant metabolic change in cancer cells is the enhancement of glycolysis. As early as 1920, Otto Warburg proposed that, compared with normal cells, most cancer cells exhibit significantly increased glucose uptake and lactic acid production, even in the presence of adequate oxygen [7]. In-depth studies have demonstrated that the abnormal proliferation of tumor cells requires sufficient energy (ATP) and specific nutrients, including nucleic acids, lipids, and proteins. Therefore, tumor cells show abnormal biosynthetic phenomena at onset and during development. Cancer cells rapidly provide ATP to support their growth via the glycolysis pathway and also furnish raw materials for the synthesis of biological macromolecules in order to promote the proliferation of tumor cells via the pentose phosphate pathway (PPP) and serine metabolism [8]. In addition, there is a mutual regulatory relationship between the metabolites produced during metabolism and by the signaling pathways in cancer cells. This means that the metabolic activities in tumors are not only regulated by oncogenes and tumor suppressor genes but also by negative feedback from their own metabolites.

### 2.1. Aberrant Biosynthesis of Cancer Cells

Proliferating cancer cells attempt to acquire more nutrients and utilize the energy derived from these substances for metabolic activities. The intermediates and products generated during the metabolic process can be used as precursor molecules for other biosynthetic reactions in tumor cells. Research studies have uncovered that the intermediate metabolites of glucose, amino acids, and fatty acids can participate in a variety of biosynthetic reactions, including the PPP, the serine synthesis pathway (SSP), and lipid synthesis, and promote the growth and proliferation of cancer cells [9]. Among these, glucose is the most important source of energy and material for tumor cells. Cancer cells not only increase the rate of glucose uptake from the extracellular environment but also accelerate their own rate of glucose metabolism to acquire more energy and intermediate metabolites. Glucose is first transported into the cell by glucose transporters (GLUT) and then metabolized via glycolysis, the tricarboxylic acid cycle (TCA cycle), and PPP pathways. However, the GLUT protein family responsible for glucose transport and several key enzymes behind glucose metabolism, such as phosphofructokinase (PFKL) and glucose-6-phosphate dehydrogenase (G6PD), is abnormally overexpressed in some specific cancer varieties [10,11,12]. It is worthy of note that the metabolism of glucose in cancer cells is an extraordinarily complex process. Studies have found that the key enzyme of the PPP pathway, G6PD, is regulated by the tumor suppressor gene *p53*; however, the *p53* gene frequently undergoes mutation in cancer cells [13,14]. Moreover, its structural homolog, the *TAP73* gene, not only activates G6PD and promotes the biosynthesis of PPP but also up-regulates the expression of PFKL via transcription, thus enhancing the Warburg effect and accelerating the proliferation of tumor cells [15]. The abnormal overexpression of glucose transporters and metabolic enzymes in cancer cells not only satisfies the energy demand required by the abnormal biosynthesis of tumors but also provides ribose 5-phosphate (R5P) for nucleic acid synthesis, and NADPH for lipid and deoxyribose biosynthesis and antioxidant defense via the PPP. These studies suggest that glucose metabolism plays an important role in the abnormal biosynthesis and proliferation of cancer cells.

In addition to glucose, tumor cells also use amino acids, such as glutamine, serine, and arginine, for energy generation, biomacromolecule synthesis, and signal transduction. These amino acids enter cancer cells via various transporters and are deployed in one-carbon metabolism and the synthesis of both nucleic acids and proteins. Among these, glutamine is extremely important for the proliferation of tumor cells as most tumor cells rely on an exogenous supply of glutamine [16]. Studies have found that the expression of alanine/serine/cysteine transporter 2 (ASCT2), which is responsible for glutamine transport, is up-regulated in many types of cancer cells, thereby providing tumor cells with the ability to absorb greater quantities of glutamine [17]. In addition, the presence of serine in tumor cells can contribute methyl groups and facilitate the one-carbon cycle. For example, intermediate 3-phosphoglyceric acid (3-PG), produced via glycolysis, is the raw material for the synthesis of serine, the process creating the appropriate conditions for the production of S-adenosylmethionine (SAM) and NADPH. These materials include phosphoglycerate dehydrogenase (PHGDH), a rate-limiting enzyme that catalyzes the oxidation of 3-PG into 3-hydroxypyruvate phosphate. In most cases, an increase in the level of PHGDH promotes the proliferation of tumor cells, whereas tumor cell proliferation is inhibited when PHGDH is knocked down or mutated at specific sites [18]. Likewise, the mono-ubiquitination of PHGDH in colorectal cancer cells increases its activity and indirectly promotes the metabolism of the one-carbon unit in cancer cells [19]. This further emphasizes the importance of serine synthesis for the proliferation of tumor cells.

In addition to the aforementioned biosynthetic responses of cancer cells, many cancer cells exhibit a high rate of de novo lipid synthesis, i.e., the synthesis of new lipids, in contrast to the preferential use of circulating lipids that characterizes most normal tissues [20]. Varies studies have found that the content of cholesterol or lipid droplets is significantly increased in various types of tumors, and that key enzymes in the fatty acid synthesis pathway, such as fatty acid synthase (FASN), acetyl-coA carboxylase (ACC), and ATP-citrate lyase (ACLY), are reactivated in different tumors, contributing to cell transformation and tumorigenesis [21]. These studies demonstrate that the growth of tumor cells is closely related to the abnormal biosynthesis of tumor cell metabolism.

### 2.2. Metabolite Signaling Molecules in Cancer Cells

Driven by the carcinogenic changes in cancer cells and the actions of host cytokines on cancer cells in the TME, cancer cells will undergo metabolic reprogramming to ensure their own growth, proliferation, and survival [22]. This means that the metabolic reprogramming of tumor cells will not only change the supply of energy and materials to the tumor; it also indicates that its metabolites will participate in the regulation of tumor-related signaling pathways (Figure 1). As mentioned earlier, cancer cells exhibit the Warburg effect. During the process of glycolysis, lactic dehydrogenase (LDH) can convert lactic acid into pyruvate. The latter product can competitively bind to proline hydroxylase 2 (PHD2) and inhibit its hydroxylation of hypoxia-inducible factor-1α (HIF-1α), thereby increasing the stability of HIF-1α. The up-regulation of HIF-1α can stimulate the expression of vascular endothelial growth factor (VEGF), promoting angiogenesis. HIF-1α can also enhance tumor progression by up-regulating the transcription of signal molecules related to antiapoptosis, invasion, and metastasis in cancer cells [23]. In addition, lactic acid, the product of glycolysis, can also directly interact with N-myc downstream-regulated gene 3 (*NDRG3*) to protect tumor cells from PHD2 hydroxylation. The accumulation of *NDRG3* has been shown to activate rapidly accelerated fibrosarcoma in the Raf/extracellular signal-regulated protein kinase (Raf/ERK) pathway to mediate angiogenesis and the proliferation of tumor cells [24]. Succinic acid is the intermediate metabolite of the TCA cycle in cancer cells, and is converted from α-ketoglutarate (α-KG) into succinic acid using succinyl-CoA and then metabolized with the use of succinate dehydrogenase (SDH) into fumaric acid. Studies have found that SDH exhibits inactivated mutations in various types of cancer cells [25,26], thus resulting in the accumulation of succinic acid in tumor cells. This can inhibit α-KG-dependent dioxygenases by competing with α-KG [27]. As with succinic acid, fumaric acid is also an intermediate metabolite of the TCA cycle. In cancer cells, fumaric acid can be catalyzed using fumarate hydratase to produce malic acid. Fumarate hydratase (FH) mutation or inactivation will lead to the accumulation of fumaric acid in cancer cells and promote the proliferation of tumor cells. For example, the deletion of FH can induce hereditary leiomyomatosis and renal cell carcinoma [28,29]. A previous in-depth study of the tumor-promoting mechanism associated with the action of fumaric acid found that the substance can also inhibit the activity of a variety of dioxygenases by competing with α-KG, thus affecting the hypoxia signaling pathway of tumor cells [30]. In addition, isocitrate dehydrogenase 1 and 2 (IDH1 and IDH2) play key roles in the metabolism of tumor cells and can convert isocitrate into α-KG. Moreover, mutations of IDH1 and IDH2 have been detected in many types of cancer, including gliomas and myeloid malignant tumors; furthermore, these enzymes have been confirmed to be closely related to the occurrence and development of cancer [31].

In addition to participating in the metabolism of tumor cells, amino acids can also promote tumor growth by regulating signaling pathways. Mammalian targets of rapamycin (mTOR) in cancer cells can regulate protein translation and participate in the regulation of tumor cell growth and autophagy. Therefore, the transmission and activation of mTOR are vital to the growth and metabolic activity of tumor cells. Previous studies found that the absence of amino acids in cell culture media could significantly inhibit the phosphorylation of downstream substrates of mTOR, thus indicating that mTOR can sense changes in amino acids [32]. Leucine and arginine can directly bind to their respective receptors, sestrin1/2 [33] and cellular arginine sensor for mTORC1 (CARTOR1) of intracellular mTOR protein complex 1 (mTORC1), thus activating mTOR and promoting protein synthesis in cancer cells [34]. Furthermore, the decomposition of glutamine can also promote the binding of GTP and RAGA/B, activating mTOR [35].

In addition to these metabolites, recent studies have revealed the important role of cholesterol in the signal transduction of tumor cells. As mentioned earlier, an increase in lipid synthesis is one of the characteristics of metabolic reprogramming in tumor cells. It is worth noting that increased lipid synthesis can lead to the accumulation of cholesterol levels in tumor cells [36]. Cholesterol can be used as a precursor in the synthesis of estrogens and androgens; these hormones can bind to their receptors and activate downstream signaling pathways, including phosphoinostitide 3-kinase (PI3K) and mitogen-activated protein kinase (MAPK), thereby promoting the proliferation of tumor cells [37,38,39]. Collectively, research has confirmed that tumor cell metabolites participate in tumor growth via the regulation of signaling molecules in tumor cells.

## 3. The Effects of Metabolic Reprogramming of Tumor Cells on Antitumor Immunity

The TME is a heterogeneous environment upon which tumor cells depend for their survival. The TME contains several vital components, including fibroblasts, immune cells, and endothelial cells, as well as various soluble secreted factors from all cellular components, such as tumor cells. These heterogeneous peripheral components are able to provide nutrients, metabolites, and signaling molecules for tumor cells; consequently, these components are vital to the proliferation and survival of cancer cells [37]. It is worth noting that nutrients such as glucose, amino acids, and lipids in the TME can be metabolized by tumor cells and immune cells via glycolysis, the TCA cycle, the PPP, amino acid metabolism, and lipid metabolism, and also provide nutrition for the rapid proliferation of cancer cells and the survival of immune cells. This means that tumor cells can induce T cell failure and immune escape by consuming vital nutrients in the TME [40]. Therefore, the mutual regulation between cancer cells and immune cells in the TME not only involves the interaction between cells but also covers the metabolic pathways responsible for the metabolic reprogramming of tumor cells and the effect of metabolites on the functionality of immune cells.

### 3.1. Tumor Cells Compete with Immune Cells for Nutrients

During the process of tumor development, abnormal metabolic changes often occur in order to meet the energy requirements for proliferation. However, metabolic transformation is not unique to cancer cells; other rapidly proliferating cells, such as activated T cells, tumor-infiltrating regulatory T cells (Tregs), and neutrophils, also conduct a large range of metabolic activities [41]. Studies have found that tumor-infiltrating lymphocytes (TILs) in the TME also require sufficient nutrients to support their proliferation and differentiation [42]. This means that the nutrients in the TME will not only be taken up by tumor cells for the purpose of meeting their own abnormal proliferation needs but will also be used by immune cells to maintain the differentiation required for antitumor activity; therefore, tumor cells and immune cells will compete for nutrients in the TME (Figure 2). Glucose is the nutrient most commonly absorbed by tumor cells, and also represents an important energy source necessary for the activation, differentiation, and function of immune cells [43]. Recent studies have found that there are differences in glucose and glutamine uptake among different cell subsets in the TME; however, this difference is not caused by the specific spatial distribution of immune cells in the TME but by the mTORC1 signal pathway and the inherent programming of glucose- and glutamine-related gene expression. Among such bodies, immune cells consume a greater amount of glucose, while tumor cells consume more glutamine [44]. This means that tumor and immune cells compete for nutrients in the TME to maintain their proliferation and functionality. For example, competitive glucose uptake by tumor cells can inhibit the function of TILs [45]. Glucose metabolism is accelerated in renal cell carcinoma (RCC) tissues expressing high levels of GLUT1, while the degree of CD8^+^ induced T cell infiltration is reduced [46]. Tumor cells can also create nutrient-deficient states for T cells by using more glutamine, thus resulting in the loss of energy, exhaustion, and even the death of T cells, and thus damaging their effector function [47]. In addition to glutamine, the expression of amino acid transporters on the surface of tumor cells and immune cells remains an important factor affecting the function of immune cells. Among these, tumor cells expressing high levels of transporters will compete with immune cells for amino acids and damage the functionality of immune cells [48]. Other studies have also confirmed that the metabolic reprogramming of tumor cells can lead to the depletion of metabolites in the TME and create a hypoxic and acidic environment, thus accelerating the competition of metabolites between invasive effector T cells and tumors and thereby damaging the functionality of effector T cells [49].

### 3.2. The Effect of Tumor Metabolic Reprogramming on Immune Cell Function

The TME contains a variety of immune cells, such as macrophages, neutrophils, monocytes, eosinophils, basophils, lymphocytes, and natural killer cells. When the body is in a steady state, these cells are quiescent; however, when the body is stimulated by infection, inflammation, or other external substances, these cells are rapidly activated and respond appropriately [50]. It is important to note that complex metabolic patterns exist not only in tumors but also in immune cells. Thus, metabolism also plays an important role in regulating immune cell phenotype and function [41,51]. During the process of tumor metabolic reprogramming, cancer cells can exert influence on immune cells; in addition, there is significant competition for nutrients. Thus, the metabolites produced by cancer cells can exert a profound impact on the growth, activation, and differentiation of immune cells in the TME.

#### 3.2.1. The Effects of Glucose Metabolism on Immune Cells

Glucose is among the primary energy sources underlying the proliferation of tumors. As mentioned earlier, glucose entering the tumor cells can be metabolized by the PPP, and G6PD is the key rate-limiting enzyme in this process. Previous studies have found that cancer cells can down-regulate the expression of G6PD via the deposition of histone H3K9 methylation (H3K9me3) on the *G6PD* promoter, and induce a decrease in the expression of granzyme B in cytotoxic T lymphocytes (CTLs), thereby promoting the proliferation of cancer cells [52]. Studies have found that, even when there is sufficient oxygen, tumor cells still tend to metabolize glucose into lactic acid. Hexokinase 2 (HK2) is a key metabolic enzyme that catalyzes the first step of the glycolysis pathway. HK2 not only plays a key role in the glycolytic metabolism of tumor cells but also performs a function that is independent of its classical metabolic function. HK2 can act as a protein kinase for the phosphorylation of nuclear factor-κB (NF-κB) inhibitor IκBα, thus causing NF-κB to enter the nucleus. This ultimately promotes the expression of PD-L1 and inhibits the activation and infiltration of CD8 T cells; collectively, these processes elevate the evasion of tumors by the immune system. The combination of an HK inhibitor and anti-PD-1 antibody was previously shown to eliminate tumor immune evasion and significantly bolster the antitumor effect of immune checkpoint blockade [53]. In addition, the glucose ingested by tumor cells can be metabolized into lactate using lactate dehydrogenase A (LDHA) during glycolytic metabolism. Other studies have found that the high expression of LDHA protein to be associated with a poor prognosis in patients with melanoma [54]. The inhibition of LDHA and interleukin-21 (IL-21) has been shown to promote the antitumor immunity of CD8^+^ T cells [55]. These data show that LDH plays a vital role in regulating the function of immune cells. It is worth noting that the glycolysis of abnormal proliferative tumor cells leads to the accumulation of large amounts of lactic acid, which is then transported out of the cell by monocarboxylate transporters (MCTs) on the cell membrane, resulting in an acidic environment in the TME [56]. Due to the ability of metabolic reprogramming, tumor cells can adapt to the adverse acidic environment in the TME. However, NK cells can be metabolically inhibited in the TME due to their lack of metabolic adaptability. Increasing the metabolic flexibility of the NK cells in the TME will improve their immunotoxic functionality and specifically enhance their antitumor activity [57]. Some studies have illustrated the positive correlation between the serum level of lactic acid in cancer patients and tumor proliferation, and further in vitro studies have confirmed that a high concentration of lactic acid in the TME can prevent the production of lactic acid in T cells, thereby interfering with their metabolism and functionality [58]. In addition, the ingestion of a large amount of lactic acid by NK cells will cause cell acidification, thus inhibiting the nuclear factor of activated T cells (NFAT); this reduces the production of NFAT-regulated interferon γ (IFN-γ) and leads to the initiation of cell apoptosis [54]. It is worth noting that the accumulation of lactic acid in the TME can act as an agonist of G protein-coupled receptor GPR81 to initiate signaling, thus promoting immune escape and cancer cell proliferation [59]. These studies have illustrated several facts. Namely, that the metabolites produced by the reprogramming of glucose metabolism in the tumor cells, or the lactic acid produced by the glycolysis pathway, can inhibit the function of immune cells, and also that cancer cells can achieve proliferation by reshaping the TME.

#### 3.2.2. The Effect of TCA Cycle Metabolism on Immune Cells

The TCA cycle is not only the primary oxidation pathway of acetyl-CoA but also the pathway that reduces nicotinamide adenine dinucleotide and flavin adenine dinucleotide. Researchers have found that the metabolites of cancer cells passing through the TCA cycle in the TME can influence the differentiation and effector function of immune cells.

For example, fumaric acid, a metabolite of the TCA in tumor cells, can enhance the immune escape ability of tumor cells by regulating CD8^+^ T cells. The loss of fumarate hydratase in tumor cells leads to the accumulation of fumarate in the TME. An increased level of fumarate can act directly on succinate kinase ZAP70 and reduce its activity in tumor-infiltrating CD8^+^ T cells, thereby inhibiting the activation of CD8^+^ T cells and the antitumor immune response, both in vitro and in vivo [60]. In contrast to the inhibitory action of fumaric acid on the activity of CD8^+^ T cells, ketoglutaric acid, an intermediate metabolite of the TCA cycle in renal cell carcinoma, is able to enhance the antitumor activity of CD8^+^ T cells by increasing the expression of major histocompatibility complex I (MHC-I) molecules in cancer cells. Furthermore, the exogenous supplementation of ketoglutaric acid in renal cancer cells has been shown to enhance the cytotoxicity of CD8^+^ T cells and reduce the escape ability of tumor cells from the immune system [61]. Succinic acid, a metabolite of the TCA cycle, can interact with toll-like receptor (TLR) ligands, induce the expression of tumor necrosis factor-α (TNF-α) and IL-1β, and activate the antigen presentation function of dendritic cells (DCs). This process can further activate T cells and thus increase the production of IFN-γ [62]. Notably, the levels of succinic acid were found to be higher in the serum of lung cancer patients when compared with serum from healthy patients. This is because cancer cells release succinate from the TCA cycle into the microenvironment; this is able to accumulate succinate receptor 1 (SUCNR1) and polarize macrophages in the tumor-associated macrophages (TAMs). Activated SUCNR1 can trigger the PI3K–HIF–1α pathway to further promote the migration, invasion, and metastasis of cancer cells [63]. In addition to these metabolites, other TCA metabolites, such as 2-hydroxyglutarate (2-HG), can also regulate the functionality of immune cells by affecting their activation and metabolism. For example, the mutation of IDH can catalyze α-ketoglutaric acid to produce the stereoisomeric and carcinogenic metabolite R-2-hydroxyglutaric (R-2-HG). The R-2-HG derived from glioma cells in the TME will be taken up by T cells; this will interfere with the transcription activity and the polyamine synthesis of the activated T cells, thereby inhibiting the activity of the T cells [64]. Similar studies have also found that IDH-mutated gliomas produce R-2-HG to promote tryptophan metabolism in infiltrating macrophages, thus leading to the activation of the aryl hydrocarbon receptor (AHR) signaling pathways in the macrophages and the inhibition of T cell function via immunosuppressive factors such as IL-10 [65]. These studies not only reveal the impact of the TCA cycle metabolites of tumor cells on immune cell activation and function but also indicate that TME remodeling is closely related to immune cell function during the metabolic reprogramming of tumor cells.

Notably, the mitochondrial TCA cycle and the electron transport chain (ETC) provide the metabolic plasticity required for cancer growth and progression. Recent studies have found that regulating an early step in mitochondrial energy production, namely allowing electron transfer primarily via complex I (CI) but not complex II (CII) of the two pathways of the ETC, can lead to the overproduction of succinate. In turn, the accumulation of succinate leads to the expression of immune genes in the nucleus, which increases the expression of the major histocompatibility complex antigen processing and presentation (MHC-APP) genes on the surface of the tumor, rendering the tumor cells more susceptible to detection and elimination by killer T cells [66]. These studies not only reveal the effects of TCA cycle metabolites and the ETC on immune cell activation and function in tumor cells but also further indicate that metabolic reprogramming of TME in tumor cells is closely related to immune cell function.

#### 3.2.3. Effects of Amino Acid Metabolism on Immune Cells

Glutamine is a substance second only to glucose in terms of the provision of energy to tumor cells. Glutamine participates in a series of biological reactions in cancer cells, including energy generation, macromolecular synthesis, and signal transduction. These mechanisms are closely related to tumor growth. In addition to their roles in tumor cells, glutamine is also vital in immune cells such as lymphocytes, macrophages, and neutrophils in terms of supporting their immune functionality. Of these, glutamine deprivation can inhibit T cell proliferation and cytokine production [67]. In addition, glutamine also possesses a high utilization rate in macrophages and neutrophils. Increasing the concentration of glutamine can reduce neutrophil apoptosis and promote the activation of macrophages and the secretion of pro-inflammatory cytokines [68]. However, T cells cultured in glutamine-free culture medium were previously shown to be unable to proliferate or produce interleukin-2 (IL-2) and IFN-γ. Furthermore, the removal of glutamine increases the levels of programmed death receptor-1 (PD-1) in cancer cells, thus resulting in the inactivation of co-cultured T cells [69,70]. Since tumor cells in the TME can undergo metabolic reprogramming, metabolic changes in the utilization of glutamine by tumor cells inevitably influence the anticancer response of immune cells. For example, recent studies have confirmed that glutamine can act as a metabolic checkpoint between tumor cells and type 1 conventional dendritic cells (cDC1s), promoting the ability of cDCs1 to activate cytotoxic T cells. The intra-tumoral supplementation of glutamine can inhibit the growth of tumor cells by enhancing cDC1-mediated CD8^+^ T cell immunity [71]. These studies strongly suggest that glutamine is essential for the functionality of immune cells. In addition to glutamine, other amino acids and their metabolites can also regulate various types of immune cell functions; furthermore, immune cells rely on amino acid metabolism to obtain energy and biological substrates [72]. For example, most tumor cells lack arginine succinic acid synthase 1 (ASS1); this manifests as a loss in the ability of cancer cells to synthesize arginine [73]. In this scenario, tumor cells use exogenous arginine to compensate for the intracellular arginine deficiency caused by a lack of key metabolic enzymes. Therefore, the abnormal proliferation of tumor cells consumes a large quantity of arginine in the TME, thus leading to arginine deficiency in the TME. While arginine metabolism plays an important role in T cell activation and the regulation of the immune response, arginine deficiency in the TME inhibits the activation of antitumor immune cells. Studies have found that exogenous arginine supplementation can stimulate T cells and NK cells to produce cytotoxic and effector cytokines and significantly enhance the antitumor immune response of immune cells [74,75]. This suggests that the metabolic reprogramming of tumor cells disrupts the metabolic balance within the TME, and that abnormal amino acid metabolism within the TME due to tumor cell metabolism interferes with the functionality of tumor-infiltrating immune cells. Therefore, inhibiting the metabolism of related amino acids and reversing the imbalance of amino acids in the TME is an effective strategy for tumor immunotherapy.

#### 3.2.4. Effects of Lipid Metabolism on Immune Cells

The abnormal proliferation of tumor cells consumes large amounts of glucose and oxygen, thus resulting in acidity, hypoxia, and glucose deficiency in the TME. These conditions force tumor cells to acquire energy via metabolic reprogramming for the purpose of maintaining proliferation [76]. In this case, lipids in the TME are activated and serve as the main energy source and key regulators for tumor cells and immune cells (Figure 3). Abnormally proliferating tumor cells must synthesize large amounts of cholesterol to protect themselves from immune surveillance [77]. For example, cholesterol-rich plasma membrane cortical soft cancer cells can evade T cell-mediated immune killing; furthermore, the depletion of cholesterol can harden the membranes of cancer cells, thereby enhancing the cytotoxicity of T cells and strengthening the efficacy of adoptive T cell therapy [78]. Furthermore, tumor cells require additional membrane structures and cholesterol synthesis during rapid proliferation; therefore, abnormally proliferating tumor cells use energy generated via glycolysis and other catabolic pathways for de novo fatty acid synthesis and the generation of plasma membrane phospholipids [79]. Notably, the increased lipid uptake and de novo cholesterol generation capacity exhibited by cancer cells can lead to the malignant transformation of tumor cells, not to mention abnormal lipid accumulation in the TME [80]. Although immune cells can undergo lipid metabolism, the abnormal accumulation of lipids in the TME will also inhibit the differentiation and antitumor ability of immune cells [75]. For example, due to the lack of key metabolic enzymes, CD8^+^ T cells cannot catabolize long-chain fatty acids accumulated within the TME, thus leading to severe lipotoxicity and T cell exhaustion [81]. These data suggest that lipids or cholesterol in the TME can provide energy to tumor cells, and that the accumulated cholesterol is also capable of regulating the functionality of immune cells. In addition, Treg cells in the TME can drive immunosuppression and mediate the immune tolerance of cancer cells. The inhibition of sterol regulatory element-binding protein (SREBP)-dependent lipid synthesis and metabolic signaling in Treg cells can release potent antitumor immune response signals without generating autoimmune toxicity [82]. This implies that lipid signaling can enhance the functional specialization of Treg cells in tumors. Similarly, a low-density lipoprotein receptor (LDLR) is a cholesterol transporter that not only participates in T cell activation and proliferation but also interacts with the T cell receptor (TCR) complex and regulates TCR circulation and signal transduction, thus promoting the effector function of cytotoxic T lymphocytes (CTLs). Furthermore, the binding of some substances in the TME to LDLRs will prevent the transfer of LDLRs and TCRs to the plasma membrane, thus inhibiting the effector function of CTLs [83]. Moreover, in vivo studies have further confirmed that high levels of serum cholesterol can increase the antitumor function of NK cells and reduce the growth of liver tumors in mice [84]. These studies suggest that in hypoxic, acidic, and nutrient-deficient TMEs, tumor cells and immune cells are more inclined to rely on lipids for energy storage and to construct signaling molecule transmission pathways between cells. However, tumor-associated immune cells in the TME can also be impacted by these ubiquitous lipid biomolecules.

## 4. Effects of Tumor-Associated Metabolic Signals on Antitumor Immunity

### 4.1. Cancer-Related Genes

The activation of oncogenes, or the inactivation of tumor suppressors, will drive the metabolic reprogramming of tumors. As the gene displaying the most frequent mutation in human tumors, *TP53* can induce apoptosis, cell cycle arrest, and cell senescence, playing an important role in tumor inhibition. The tumor-related mutant p53 protein cannot exert the antitumor functionality of wild-type p53, and acquires a new carcinogenic function that is independent of wild-type p53 [85]. Mutant p53 has been shown to disrupt the cytoplasmic DNA-sensing mechanism that activates the innate immune response, cGAS-STING-TBK1-IRF3, by binding to TANK. This binds to protein kinase 1 (TBK1) and prevents the formation of a trimeric complex between TBK1, stimulator of interferon genes (*STING*), and the transcription factor IRF3. Thus, the immune surveillance function of the immune cells is inhibited, and tumor growth is accelerated [86]. Similar studies have also found that mutant p53 reduces the activation of T cells via the transcriptional inhibition of pleckstrin homology domain leucine-rich repeat protein phosphatase 2 (PHLPP2) and the activation of protein kinase B (AKT), thereby promoting the immune escape and growth of tumor cells [87]. In addition, carcinogenic mutations in the *KRAS* gene can also promote tumor immune evasion, survival, and growth in colorectal cancer, non-small cell lung cancer, and pancreatic ductal adenocarcinoma [88]. In addition, the oncogene *MYC* has also been reported to up-regulate the DNA methylation gene *DNMT1* in triple-negative breast cancer. This inhibits STING-dependent innate immunity, thus inducing the escape of cancer cells from the immune system [89]. These studies show that cancer cell-related genes associated with the process of tumorigenesis and development can affect the activation and function of immune cells.

### 4.2. Cancer-Related Signaling Pathways

Metabolic reprogramming not only occurs in cancer cells but can also be observed in activated T cells, activated B cells, activated NK cells, and dendritic cells [90]. Immune cells are exposed to various metabolic states during their activation and self-renewal. Indeed, in general, naïve T cells exhibit a low level of metabolite demand. However, once T cell receptors recognize antigens and receive co-stimulatory signals from antigen-presenting cells, the metabolic profile of T cells changes [91]. During activation, cells shift the T cell metabolism from fatty acid β-oxidation and pyruvate oxidation to aerobic glycolysis and PPP via the TCA cycle [92]. Additionally, solute carrier family 2 member 1 (SLC2A1), which is responsible for glucose entry into the cell, is rapidly up-regulated upon T cell and macrophage activation [93]. Thus, it helps activated T cells to participate in high-rate aerobic glycolysis in terms of enhancing proliferation and effector functions.

The metabolic reprogramming of tumor cells is not only regulated by oncogenes but is also precisely regulated by oncogenic signaling molecules and related metabolic signaling pathways. In cancer cells, mTOR can regulate protein translation and participate in the regulation of growth and autophagy in tumor cells. Therefore, the transmission and activation of mTOR signaling is extremely important for the growth and metabolic activities of tumor cells. However, recent studies have found that the mTOR signaling pathway not only directly affects the energy metabolism of tumor cells but also plays a role by influencing the antitumor activity of immune cells. For example, the lactic acid produced via glycolysis in cancer cells can activate the mTOR pathway and inhibit the expression of macrophage-specific ATP subunit ATP6V0d2, mediated by transcription factor TFEB, thus resulting in increased HIF-2 α-mediated VEGF production in the macrophages. This, in turn, promotes the growth of tumor cells [94]. In addition to lactic acid, amino acids in the TME can also directly activate mTORC1 in CD8^+^ T cells via amino acid sensors and confer antitumor immune activity to T cells [95]. Notably, the upstream effector molecule of mTOR, AMP-activated protein kinase (AMPK), is a key metabolic molecule that induces mitochondrial biogenesis. Studies have found that energy deprivation can activate AMPK to enhance the antitumor activity of immune cells [96]. In T cells, AMPK activation involves a variety of functions, such as the inhibition of T cell response, the promotion of T cell regulatory function, and the production of CD8^+^ memory T cells [97]. This means that various factors in the TME, such as nutritional deprivation, adenosine, and anti-inflammatory cytokines, can induce the activation of AMPK in tumor-associated macrophages (TAMs) and infiltrating T cells. This process results in alterations in intracellular metabolic signaling pathways, thereby inhibiting antitumor immune activity and promoting tumor growth.

In addition to the mTOR pathway, other substances in the TME can also regulate the antitumor activity of immune cells via certain signal pathways. For example, asparagine (Asn), which exists in the TME, not only modulates the proliferation and growth of tumor cells but also plays an important role in the activation of immune cells [98]. Studies have found that Asn binds to SRC family lymphocyte-specific protein tyrosine kinase (LCK) and coordinates the phosphorylation of LCK at Tyr 394 and 505, thus leading to enhanced LCK activity and T cell receptor signaling, thereby improving the antitumor response of CD8^+^ T cells [99]. In addition, the intracellular level of nicotinamide adenine dinucleotide (NAD^+^) is also essential to the functionality of immune cells. In TILs, the low expression of nicotinamide phosphoribosyltransferase (NAMPT), controlled by the tubby (TUB), has been shown to impair the NAMPT-dependent NAD^+^ salvage pathway, thus leading to NAD^+^ depletion in cells. However, the reduction in NAD^+^ disrupts the TCR–TUB–NAMPT–NAD^+^ signaling pathway in TIL cells, thus leading to T cell dysfunction in the TME. Studies involving a mouse model further showed that exogenous NAD^+^ could enhance the tumor-killing efficacy of T cells [100]. These studies show that cancer-related signaling pathways not only regulate the growth of tumor cells but also participate in the regulation of antitumor activity in immune cells.

## 5. Impact of the Tumor Metabolic Microenvironment on CAR T Cell Therapy

Chimeric antigen receptor (CAR) T cell therapy marks a new era in cancer immunotherapy during tumor treatment. CAR T cells are engineered T cells that deploy recombinant receptors for target antigen recognition and exert an effective antitumor response in a major histocompatibility complex (MHC)-unrestricted manner [101]. However, the efficacy of CAR T cell therapy in the treatment of solid tumors is largely limited. This is primarily due to the fact that TMEs in solid tumors are more complex, with physical barriers, multiple immunosuppressive mechanisms, and a variety of cancer metabolites limiting the efficacy of CAR T cell therapy. During the therapeutic process, the fate of T cell-infiltrating TMEs following binding to the tumor target is closely related to the metabolic reprogramming occurring within the tumor [102].

As mentioned earlier, tumor cells need to consume large amounts of glucose for energy during their abnormal proliferation. It has been found that the consumption of large quantities of glucose inhibits T cell glycolysis and the production of the downstream metabolite phosphoenolpyruvate (PEP). The latter is a key metabolite in the regulation of antitumor responses. T cells that overexpress phosphoenolpyruvate carboxykinase 1 (PCK1) increase PEP production, limit tumor growth, and prolong the survival of melanoma mice [103]. Since tumor cells prefer to metabolize glucose via the glycolytic pathway, there is a high accumulation of lactate within the TME. The presence of high lactate levels in the TME is usually associated with lower numbers and reduced activity of CD8^+^ T cells and NK cells in vitro and in vivo [54]. It has been found that high lactate levels inhibit T cell functions, including interleukin-2 secretion and T cell receptor activation, while blocking lactate dehydrogenase A in tumor cells improves immune function and the efficacy of anti-programmed cell death 1 therapy [104]. Similarly, other studies have demonstrated that the knockdown of lactate dehydrogenase A in tumor cells improves the efficacy of CAR T cell therapy and significantly reduces tumor growth [105]. In addition, the level of amino acids in the TME exerts an equally large impact on CAR T cell function. For example, low levels of arginine in the TME impair the function of CAR T cells lacking expression of argininosuccinate synthase (ASS) and ornithine transcarbamylase (OTC). Thus, enzyme-modified CARs based on this design, expressing functional ASS or OTC, was shown to be effective in increasing CAR T cell proliferation and improving solid tumor clearance without affecting the degree of cytotoxicity or exhaustion of the CAR T cells [103]. These studies suggest that the onset of metabolic reprogramming of the tumor cells impacts the TME and further hinders the development of tumor-immune CAR T therapies. Continuously improving the persistence of CAR T cells and overcoming resistance to TME metabolic stress are key challenges for CAR T cell therapy.

## 6. Tumor Immunotherapy Based on Metabolic Regulation

During tumor development, tumor cells not only meet their own proliferation needs via metabolic reprogramming but also remodel the TME using the metabolites or cytokines produced via metabolism. These latter effects limit the functionality of the immune cells and provide obstacles for cancer treatment [106]. Therefore, targeting the metabolic pathways of tumor cells and immune cells to improve the antitumor activity of immune cells or overcome tumor immune escape via immune checkpoint blockade is an effective treatment strategy for tumor immunotherapy (Table 1).

As mentioned earlier, the massive proliferation of tumor cells requires the release of energy from glucose. Glucose analog 2-deoxy-D-glucose (2DG) targets the glucose metabolism to deplete energy from cancer cells. In addition, 2DG increases oxidative stress, inhibits N-linked glycosylation, and induces autophagy. It effectively slows cell growth and efficiently promotes apoptosis in specific cancer cells [107]. Treatment with 2DG disrupts N-glycan coverage on tumor cells and leads to enhanced CAR T cell activity in different xenograft mouse models of pancreatic cancer. In addition, 2DG treatment interferes with the PD-1–PD-L1 axis and leads to a reduced depletion profile of tumor-infiltrating CAR T cells in vivo [108]. This suggests that the combination of 2DG and CAR T cell therapy is valuable for the treatment of tumors. In tumor cells, glycolysis produces lactate, thus leading to increased acidity in the TME and the impairment of immune cell function. As such, neutralizing the low pH in the TME may exert a meaningful impact in terms of improving the efficacy and outcomes of anticancer immunotherapy. For example, a super-carbonate-buffered TME via oral administration inhibits tumor growth when combined with anti-PD-1 immunotherapy in melanoma models, thus improving patient survival when combined with adoptive T cell immunotherapy [109,110]. In addition, the concentration of lactic acid in the TME can be regulated by inhibiting the lactic acid transporter. For example, the inhibition of monocarboxylic acid transporter 1/4 (MCT 1/4) can effectively increase the lactic acid in tumor cells, reduce the rate of glycolysis, and accelerate the death of cancer cells [111]. It has been reported that this treatment can enhance the secretion of IL-2 and IFN-γ in T cells [112]. Among these, AZD3965, an inhibitor of the monocarboxylic acid transporter family 1 (MCT1), is in early clinical trials (http://www.clinicaltrials.gov/ (accessed on 30 November 2023)). The MCT family comprises transmembrane proteins that mediate the transportation of lactate in and out of the cell by tumor cells, and play a key role in regulating lactate homeostasis in tumors [113]. It was found that an AZD3956 blockade of MCT1 could not only inhibit lactate transport and lipid biosynthesis in cancer cells but could also regulate the infiltration of tumor immune cells in vivo. In particular, they were deemed especially capable of increasing the abundance and maturation of NK and DC cells in tumors and improving the efficacy of CAR T cell therapy for B cells [114,115]. This means that blocking MCT can inhibit the proliferation of tumor cells and support the activation of immune cells. In addition to glucose, glutamine is the most abundant amino acid in the metabolic cycle and is also the nutrient consumed the fastest by cancer cells [16]. The inhibition of glutaminase by CB-839 not only limits tumor cell growth but also can be combined with anti-PD-1 immunotherapy to promote the glucose metabolism, epigenetic reprogramming, and cytotoxicity functions within T cells, improving the antitumor effects of immune cells [116]. Similarly, JHU083, a prodrug of 6-diazo-5-oxo-L-norleucine (L-DON), is a small-molecule drug that targets the glutamine metabolism [117]. Some research has found that a JHU083 blockade of glutamine metabolism significantly inhibited the production and recruitment of myeloid-derived suppressor cells (MDSCs) and promoted the production of antitumor inflammation-associated macrophages. Surprisingly, JHU083 also inhibited indoleamine 2,3-dioxygenase (IDO) expression in both tumor and myeloid-derived cells, leading to a significant reduction in kynurenine levels. This in turn inhibited the development of metastasis and further enhanced the antitumor immunity [118]. IDO1 is an intracellular immunomodulatory enzyme that promotes immunosuppression, tolerance, and tumor escape by catabolizing tryptophan. This is the first and rate-limiting step in the degradation of tryptophan, leading to the subsequent production of kynurenine. The overexpression of IDO1 has been associated with poor patient survival in several types of human cancers [119]. INCB024360 is an orally reversible, competitive, and potent IDO1 inhibitor in clinical studies [120]. The treatment of DCs with this inhibitor increases their ability to stimulate CD8 antigen-specific T cell lines in vitro and increases tumor cell lysis using antigen-specific CD8^+^ T cell lines derived from cancer patients [120]. In addition, adenosine, which is released in large quantities by cancer cells in the tumor microenvironment, is among the primary immunosuppressive agents responsible for the escape of cancer cells from immune surveillance [121]. The administration of SCH58261, an inhibitor of the A2A adenosine receptor, reawakened T cell responses while limiting Treg amplification and repolarizing monocytes to an inflammatory (M1-like) phenotype in a mouse model of chronic lymphocytic leukemia [121]. Similarly, CPI-44 selectively and efficiently binds to the A2A receptor and competitively inhibits adenosine binding and signaling. This molecule was found to safely block adenosine signaling in vivo in a phase 1 clinical trial and was used for immunotherapy to treat refractory renal cell carcinoma [122].

Recent studies have found that the combination of glutamine metabolism blockade and anti-PD-1 immunotherapy will not only inhibit the metabolism of tumor cells but also promotes glucose metabolism, epigenetic reprogramming, and cytotoxic function in T cells to improve the antitumor effect of immune cells [116]. Collectively, these studies indicate that the reprogramming of metabolites and the TME play important roles in regulating the functionality of immune cells; this strategy has been considered an attractive target for cancer immunotherapy.

**Table 1 ijms-24-17422-t001:** Tumor immunotherapy drugs targeting metabolic regulation.

Drug	Target	Mechanism	References
AZD3965	Monocarboxylate transporter 1	potentiates NK cell activity	[114]
CB-839	Glutaminase	promotes the differentiation of CD4^+^ T-helper 1 cells and CD8^+^ cytotoxic T lymphocytes	[123]
JHU083	Glutamine antagonists	induces the differentiation of MDSCs and TAMs into pro-inflammatory TAMs	[118]
L-DON	Glutamine-fructose amidotransferase 1	enhances CD8^+^ T cell infiltration into tumors	[124]
2DG	Hexokinase	promotes the development of naïve T cells into Treg cells	[125]
INCB024360	Indoleamine 2,3-dioxygenase	induces T/NK cell proliferation	[119,120]
SCH58261	Adenosine	enhances CD8^+^ T cells response and decreases regulatory T cells	[126]
CPI-444	Adenosine A2a receptor	modulates expression of T cell coinhibitory receptors and improves effector function	[127]

## 7. Discussion

Abnormal energy metabolism is among the most important markers of malignant tumors, whereby tumor cells provide energy for the rapid proliferation of cancer cells via metabolic reprogramming. Increasing evidence suggests that cancer metabolism not only plays a key role in maintaining signals for tumor initiation and progression but also remodels the TME to maintain an environment that is conducive to tumor proliferation by releasing metabolites and influencing the expression of immune molecules. In addition, the energetic interaction between tumor cells and immune cells in the TME can lead to a competitive phenomenon in the metabolism of the tumor ecosystem. This limits the availability of nutrients, thus leading to acidosis of the microenvironment and ultimately hindering the functionality of immune cells. More interestingly, immune cells also undergo metabolic reprogramming during proliferation, differentiation, and the execution of effector functions that are essential for immune responses. Therefore, during tumor growth, it is not only the case that the metabolic programming of tumor cells affects the antigen presentation and recognition of immune cells. Rather, the metabolic programming of immune cells also affects their own functionality, eventually leading to changes in tumor immunity.

The interaction between immune cells and cancer cells is a complex and dynamic process, and immune cells are the decisive factor in the fate of cancer cells. It is worth noting that the metabolic interaction between the immune cells and cancer cells in the TME is not only a key feature of tumor biology but also constitutes a key breakthrough in the discovery of cancer cell vulnerability. In view of the important role of immune cells in cancer progression, it is vitally important to understand the effect of tumor metabolic reprogramming on immune cell function in the TME and the ways in which it affects the progression of cancer. As mentioned earlier, the process of metabolic reprogramming in tumor cells affects antigen presentation and recognition by immune cells, thereby mediating the immune escape process of cancer cells. Therefore, by exploiting the metabolic crosstalk between tumor cells and immune cells, it is possible to improve the efficiency of tumor immunotherapy and target metabolites or metabolic enzymes to interfere with the metabolic reprogramming of tumor cells. This can regulate immune cells in the TME and, as such, is more conducive with regards to antitumor immune function.

Although various combinations of tumor metabolic inhibitors and immunotherapies have been applied in clinical trials, the complexity of the relationship between tumor cell metabolic reprogramming and the TME means that a single therapeutic target is not sufficient to block the tumor-promoting connection between cancer cells and the surrounding immune cells. Therefore, it is necessary to understand the metabolic mechanisms of tumor immune evasion and the metabolic demands of immune cells in the future, and also to identify therapeutic targets that interrupt the tumor-promoting relationship between cancer cells and the TME, in order to maximize the efficacy of immune checkpoint inhibitors and other gene and cell therapies.

## Figures and Tables

**Figure 1 ijms-24-17422-f001:**
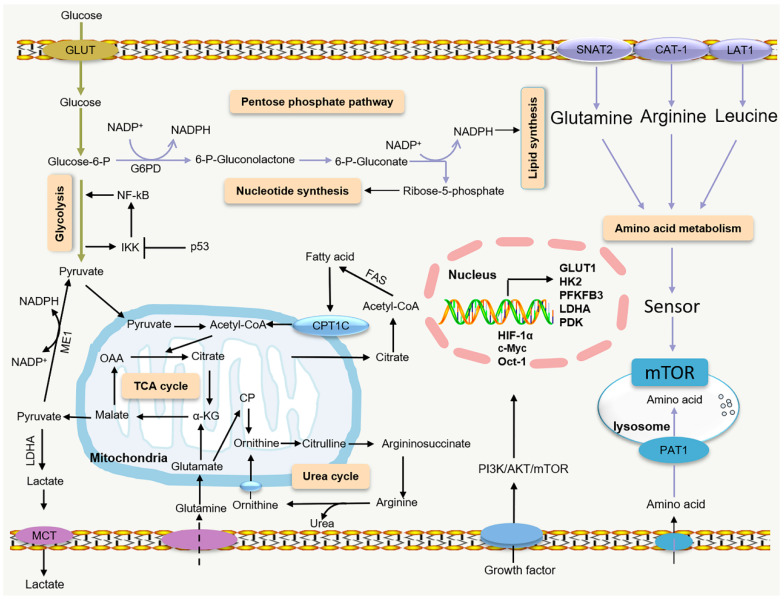
Metabolic and signaling pathways supporting tumor biomass production. PI3K–AKT–mTORC1 pathways are often deregulated in cancer cells and increase flux via glycolysis, the tricarboxylic acid (TCA) cycle, and amino acid metabolism to support tumor progression. GLUT: glucose transporter; Glucose-6-P: glucose 6-phosphate; G6PD: glucose-6-phosphate dehydrogenase; NF-kB: nuclear factor-k-gene binding; IKK: Ikappa B kinase; ME1: malic enzyme 1; LDHA: lactate dehydrogenase A; MCT: monocarboxylate transporter; OAA: oxaloacetate; α-KG: α-ketoglutaric acid; CP: carbamoyl phosphate; CPT1C: carnitine palmitoyl transferase 1C; FAS: fatty acid synthase; HK2: hexokinase 2; PFKFB3: fructose-2,6-bisphosphatase 3; PDK: pyruvate dehydrogenase kinases; SNAT2: sodium-dependent neutral amino acid transporter 2; CAT-1: cationic amino acid transporter 1; LAT1: L-type amino acid transporter 1; PAT1: proton-coupled amino acid transporter 1.

**Figure 2 ijms-24-17422-f002:**
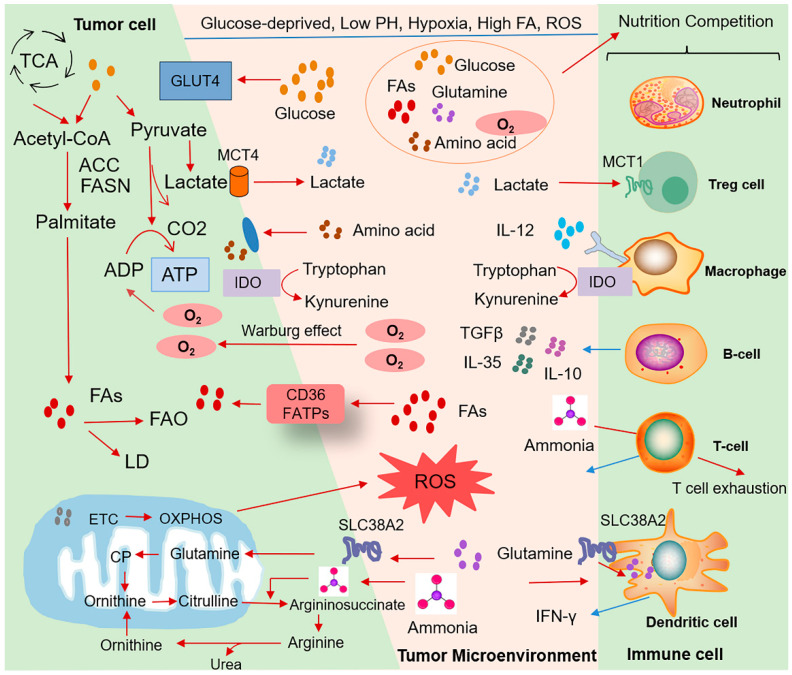
Tumor cells compete with immune cells for nutrients. The nutritional competition between tumor cells and immune cells inside tumors. The competition-caused deficiency of glucose and its metabolite lactate can affect the function of immune cells, including Tregs, macrophages, dendritic cells, natural killer (NK) cells, and so on. The extensive crosstalk between cancer cells and their microenvironment, including immune cells, and the extracellular matrix facilitates cancer progress via reciprocal metabolic and immune regulation. TCA: tricarboxylic acid cycle; ACC: acetyl CoA carboxylase; FASN: fatty acid synthase; FAs: fatty acids; FAO: fatty acid oxidation; LD: lipid droplet; ETC: electron transport chain; CP: carbamoyl phosphate; OXPHOS: oxidative phosphorylation; GLUT: glucose transporter; MCT: monocarboxylate transporters; IDO: indoleamine 2,3-dioxygenase; FATPs: fatty acid translocase.

**Figure 3 ijms-24-17422-f003:**
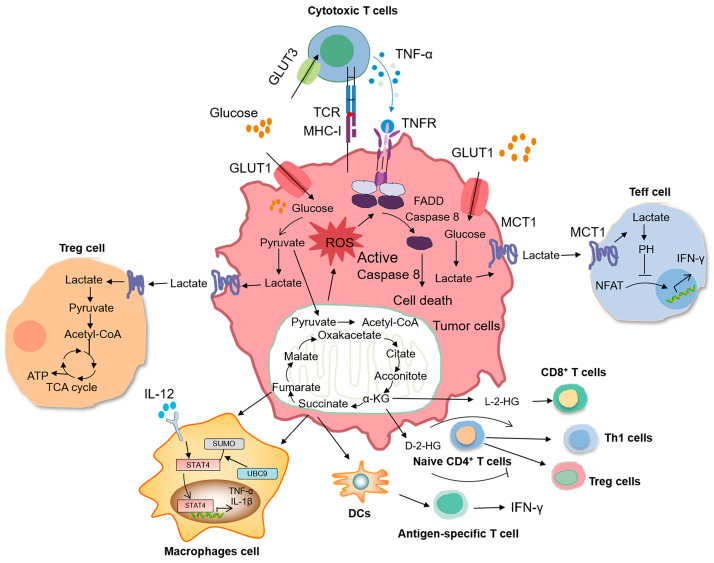
Effect of tumor metabolic reprogramming on immune cell function. GLUT1 deficiency leads to a significant increase in mitochondrial oxidative phosphorylation (OXPHOS) levels and the production of a large amount of reactive oxygen species (ROS). The accumulation of ROS in cells leads to a down-regulation of c-Flip levels, thereby exacerbating TNF-α induced cell apoptosis. The lactic acid produced by cancer cell proliferation and metabolism is discharged into the TME, and some immune cells, including Treg cells and Teff cells, can sense the level of lactic acid, triggering intracellular signals, fine-tuning cell behavior, and strongly affecting their function. In addition, TCA cycle metabolites within tumor cells, such as fumarate, succinate, and α-metabolites such as KG, can affect the differentiation of macrophages and T cells. ROS: reactive oxygen species; TCR: T cell receptor; MHC-I: major histocompatibility complex I; FADD: Fas-associating protein with a novel death domain; NFAT: nuclear factor of activated T cells.

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
