# Peer review of "The Role of Tumor Metabolic Reprogramming in Tumor Immunity"

_ijms, 2023, doi:10.3390/ijms242417422_

Round 1
Reviewer 1 Report
Comments and Suggestions for Authors
The article of Zhang et al. on ‘The role of tumor metabolic reprogramming in tumor immunity’ reviews the current literature focusing on tumor metabolism and metabolic interactions of cancer cells with immune cells in the tumor microenvironment. The authors gather largely established information from important works during the last two decades and summarize key concepts in six chapters, including subchapters as well as three overview figures and one table. The topic and basic outline of this review is not in itself novel and the discussed topic has been reviewed comprehensively and in greater detail before (e.g. Arner & Rathmell, 2023, Cancer Cell and others). However, the authors produced solid work, which may fit the scope and audience of the journal and could be accepted for publication after addressing the following points:
- While the article is largely written in a comprehensible manner, some passages are of poor language, appear repetitive and are stylistically meager. To address this, I suggest English language editing. In addition, the authors should double-check generalizing statements with respect to scientific validity (e.g. the authors use the phrase ‘occurrence and development of tumors’ throughout their work, while I am not sure it is adequate in every case.)
- The authors should include some further information whether mutations in metabolically relevant enzymes can affect patient survival or therapeutic efficacy. Also, the effects of metabolites in the TME in e.g. CAR T cells therapy should be discussed.
- In chapter 2.2 the authors state ‘… accelerated fibrosarcoma in the rats/extracellular signal-regulated protein kinases (RAF/ERK) pathway…’, while RAF in fact abbreviates for 'rapidly accelerated fibrosarcoma'.
- The style and layout of all three figure is very much different. I strongly suggest to harmonize styles, abbreviations, density and details of information in the figure for the sake of readability.
- Chapter 3.2.1: The sentence ‘… and induce the depletion of cytotoxic T lymphocyte (CTL) to reduce the expression of granzyme B, thus promoting their own proliferation [52].’, needs to be revised
- Last sentence 3.2.1 ‘…; thus, cancer cells can undergo proliferation following TME remodeling.’, needs to be revised.
- Figure 2 lacks abbreviations in the legend.
- Figure 2 contains the non-official/faulty protein names IF-12 and IF-gamma. Please correct.
- In 3.2.2: ‘R-2-HG derived from glioma cells in the TME will be taken up by T cells; this will interfere with the transcription activity of nuclear factor and the polyamine synthesis of activated T cells, thereby inhibiting the activity of T cells [64].’ The authors certainly mean nuclear factor of activated T cells (NFAT).
- In 3.2.4 ‘For example, cholesterol-rich plasma membrane cortical soft cancer cells are resistant to T-cell-mediated immune killing;’, needs to be revised.
- In table 1, the authors only mention the tumor immunotherapy drugs targeting metabolic regulation, but do not elaborate on the reports/trials. It would be interesting to read more on the single studies.
Comments on the Quality of English Language
I suggest some English language editing
Author Response
The article of Zhang et al. on ‘The role of tumor metabolic reprogramming in tumor immunity’ reviews the current literature focusing on tumor metabolism and metabolic interactions of cancer cells with immune cells in the tumor microenvironment. The authors gather largely established information from important works during the last two decades and summarize key concepts in six chapters, including subchapters as well as three overview figures and one table. The topic and basic outline of this review is not in itself novel and the discussed topic has been reviewed comprehensively and in greater detail before (e.g. Arner & Rathmell, 2023, Cancer Cell and others). However, the authors produced solid work, which may fit the scope and audience of the journal and could be accepted for publication after addressing the following points:
While the article is largely written in a comprehensible manner, some passages are of poor language, appear repetitive and are stylistically meager. To address this, I suggest English language editing. In addition, the authors should double-check generalizing statements with respect to scientific validity (e.g. the authors use the phrase ‘occurrence and development of tumors’ throughout their work, while I am not sure it is adequate in every case.)
Author's Response: Thank you very much for your suggestions, we have re-edited the full text in English and hopefully improved the style and fluency of the article. In addition, we have revised the phrase "tumorigenesis and progression" in the manuscript and ensured that it does not appear in inappropriate cases.
The authors should include some further information whether mutations in metabolically relevant enzymes can affect patient survival or therapeutic efficacy. Also, the effects of metabolites in the TME in e.g. CAR T cells therapy should be discussed.
Author's Response: Thank you very much for your suggestion, although some of the metabolizing enzyme mutations have been included in the article (e.g. the sentence "Moreover, mutations of IDH1 and IDH2 have been detected in many types of cancers, including gliomas and myeloid malignant tumors"), but does not contain further information on whether the relevant metabolizing enzymes affect patient survival or treatment outcomes. This is because the effect on patient survival or treatment outcome following metabolic enzyme mutations is more likely to result from metabolic enzyme mutations altering metabolites or metabolic signaling pathways, and the mechanisms in this regard have been less well studied and are not the focus of what we would like to address. In addition, due to the importance of metabolites in TME in CAR-T cell therapy, we have summarized them in a separate overview and added Chapter 5.
In chapter 2.2 the authors state ‘… accelerated fibrosarcoma in the rats/extracellular signal-regulated protein kinases (RAF/ERK) pathway…’, while RAF in fact abbreviates for 'rapidly accelerated fibrosarcoma'.
Author's Response: Thank you for your assistance, it has been modified.
The style and layout of all three figure is very much different. I strongly suggest to harmonize styles, abbreviations, density and details of information in the figure for the sake of readability.
Author's Response: Thank you for your suggestions. Due to the need to highlight the inter-crosstalk between various signaling pathways, metabolites and immune cells and to ensure the rigor of the scientific issues in the figure, the authors have modified only some of the abbreviations, densities.
Chapter 3.2.1: The sentence ‘… and induce the depletion of cytotoxic T lymphocyte (CTL) to reduce the expression of granzyme B, thus promoting their own proliferation [52].’, needs to be revised.
Author's Response: Thank you for your assistance, it has been modified.
Last sentence 3.2.1 ‘…; thus, cancer cells can undergo proliferation following TME remodeling.’, needs to be revised.
Author's Response: Thank you for your assistance, it has been modified.
Figure 2 lacks abbreviations in the legend.
Author's Response: Thank you for your assistance, it has been modified
Figure 2 contains the non-official/faulty protein names IF-12 and IF-gamma. Please correct.
Author's Response: Thank you for your assistance, it has been modified
In 3.2.2: ‘R-2-HG derived from glioma cells in the TME will be taken up by T cells; this will interfere with the transcription activity of nuclear factor and the polyamine synthesis of activated T cells, thereby inhibiting the activity of T cells [64].’ The authors certainly mean nuclear factor of activated T cells (NFAT).
Author's Response: Thank you for your assistance, it has been modified
In 3.2.4 ‘For example, cholesterol-rich plasma membrane cortical soft cancer cells are resistant to T-cell-mediated immune killing;’, needs to be revised.
Author's Response: Thank you for your assistance, it has been modified
In table 1, the authors only mention the tumor immunotherapy drugs targeting metabolic regulation, but do not elaborate on the reports/trials. It would be interesting to read more on the single studies.
Author's Response: Thank you for your suggestions. Due to limitations in table style and content, the authors have provided a detailed description of the mechanism of action of various drugs and related experiments in Chapter 6.
Reviewer 2 Report
Comments and Suggestions for Authors
Solid tumors develop and progress while being embedded in a complex tumor micro-environment (TME), which is composed of a diverse repertoire of cells. Under this frame, cancer cells reprogram their metabolic systems thereby enabling and sustaining their abnormal and de-regulated growth. However, the implemented metabolic outcomes affect not only the malignant cells themselves, but other cellular constituents of the TME, as well. Of note are immune cells which reside in the TME and can take part in the currently intensively pursued, immune-checkpoint inhibitors (ICI), therapeutic approaches. Reviewing and discussing the interplay and mutual effects between cancer and immune cells in the TME, are therefore of profound importance, and most relevant to the further improvement of cancer immuno-therapy. Along this line, the authors focus in the current review on the role of tumor metabolic reprogramming in tumor immunity. While this review is detailed and comprehensive, it lacks in some aspects, depth and clarity. For example, when discussing “Cancer related signaling pathways” and their effects on immune-cells (chapter 4.2), the authors should first briefly describe the various metabolic states that accompany the activation and self-renewal stages of immune T-cells. This is imperative for clearly understanding the processes discussed in this section, and their impact on the TME immune cells. Furthermore, it should be noted that some statements in this section are inaccurate. For example, the authors state that AMPK activates the oxidative-phosphorylation (OxPhos) process, thereby down-regulating and suppressing T-cells. From the relevant literature it emerges that AMPK is not inducing OxPhos but is rather inducing mitochondrial biogenesis which by itself is expected to support the self-renewal of T-cells. Although quiescent T-cells mainly use OxPhos to generate ATP, and upon activation they upregulate glycolysis, OxPhos is essential for the self-renewal of the activated cells. Collectively, I would argue that AMPK can suppress T-cells through the down-regulation of mTORC1 and not through the induction of the OxPhos.
In addition, in the section discussing the effect of cancer cells TCA cycle on the immune-cells activity, the authors should refer also to the recent finding that succinate accumulation in electron transport chain inhibited cancer cells, induces antigens presentation on the outer membrane of the treated cells. This obviously potentiates the immunogenicity of the treated cancer cells.
After revising the review according to the said above, this review can be accepted for publication.
Comments on the Quality of English Language
The manuscript could benefit from some additional professional editing.
Author Response
Solid tumors develop and progress while being embedded in a complex tumor micro-environment (TME), which is composed of a diverse repertoire of cells. Under this frame, cancer cells reprogram their metabolic systems thereby enabling and sustaining their abnormal and de-regulated growth. However, the implemented metabolic outcomes affect not only the malignant cells themselves, but other cellular constituents of the TME, as well. Of note are immune cells which reside in the TME and can take part in the currently intensively pursued, immune-checkpoint inhibitors (ICI), therapeutic approaches. Reviewing and discussing the interplay and mutual effects between cancer and immune cells in the TME, are therefore of profound importance, and most relevant to the further improvement of cancer immuno-therapy. Along this line, the authors focus in the current review on the role of tumor metabolic reprogramming in tumor immunity. While this review is detailed and comprehensive, it lacks in some aspects, depth and clarity.
For example, when discussing “Cancer related signaling pathways” and their effects on immune-cells (chapter 4.2), the authors should first briefly describe the various metabolic states that accompany the activation and self-renewal stages of immune T-cells. This is imperative for clearly understanding the processes discussed in this section, and their impact on the TME immune cells. Furthermore, it should be noted that some statements in this section are inaccurate. For example, the authors state that AMPK activates the oxidative-phosphorylation (OxPhos) process, thereby down-regulating and suppressing T-cells. From the relevant literature it emerges that AMPK is not inducing OxPhos but is rather inducing mitochondrial biogenesis which by itself is expected to support the self-renewal of T-cells. Although quiescent T-cells mainly use OxPhos to generate ATP, and upon activation they upregulate glycolysis, OxPhos is essential for the self-renewal of the activated cells. Collectively, I would argue that AMPK can suppress T-cells through the down-regulation of mTORC1 and not through the induction of the OxPhos.
Author's Response: Thanks to your suggestion, the authors have added to the text the various metabolic states that accompany the activation and self-renewal phases of immune t cells. After discussion, the authors have also concluded that AMPK can inhibit t-cells by down-regulating mTORC1 rather than by inducing OxPhos, and have therefore changed some of the statements in this section.
In addition, in the section discussing the effect of cancer cells TCA cycle on the immune-cells activity, the authors should refer also to the recent finding that succinate accumulation in electron transport chain inhibited cancer cells, induces antigens presentation on the outer membrane of the treated cells. This obviously potentiates the immunogenicity of the treated cancer cells.
Author's Response: Thanks to your suggestion, the authors have referenced recent work on the accumulation of succinate in the electron transport chain to inhibit cancer cells in the TCA cycle section.
After revising the review according to the said above, this review can be accepted for publication.
Reviewer 3 Report
Comments and Suggestions for Authors
In this work Zhang and colleagues explore the metabolic reprogramming occurring during tumor development in the tumor microenvironment and its impact on the survival and function of immune cells. The topic is original considering that it includes 3 relevant topics of great interest nowadays: the impact of metabolic reprogramming in 1. the tumor microenvironment; 2. Cancer development and 3. immune cell survival and function. Despite being a very complex topic, the author's proposal contributes to this developing field.
In comparison to other reviews this manuscript integrates a more complex and physiological panorama, as most of the current reviews in these areas only explore a part (isolate) the topics e.g, the effect of tumors in the TME; or the interplay between the immune system and the tumor progression; the effect of metabolic reprogramming in cancer cells… but very few original works discuss the interaction between all these components. And I think that is a strength of the present manuscript.
However, despite the manuscript is adequate in the present form many aspects must be attended in the manuscript in order to improve the quality of this work and improve the audience appreciation
Many observations mentioned by the authors as “nutrient deprivation”, “absence of amino acids in the cell culture” can promote an autophagic panorama. Alterations in autophagy (induction or inhibition) is another way to modulate cancer metabolism. Extensive and recent reviews are available regarding the role of autophagy in cancer cell metabolism, in the present manuscript the authors hardly (2 times) mention autophagy. The authors may want to extend and reinforce this idea.
In abstract: “The occurrence and development of tumors require the metabolic reprogramming of cancer cells. Cancer cells can change their …” rephrase is encouraged, as it is repetitive in the present form.
In abstract: “antitumor” is employed while in other parts of the text anti-tumor, please homogenize.
In introduction: “Previous studies have found that the occurrence and development of tumors are closely related to the tumor microenvironment (TME). The abnormal accumulation of metabolites produced by the massive proliferation of tumor cells can promote the occurrence and development of tumors” rephrase to avoid repetitive text.
Page 3: de novo, and other Latinism as in vivo, in vitro, etc., must be italicized
Page 3: “convert lactic acid to pyruvate. Pyruvate can competitively bind to proline hydroxylase 2 ” IDEM, rephrase, perhaps as: to pyruvate. The latter can…
Page 3: downstream-regulated gene 3 (NDRG3) and other genes must be italicized.
Page 4: “Thus resulting in the accumulation of succinic acid in tumor cells. The accumulation of succinic acid” IDEM, rephrase please.
Page 4: “action of fumaric acid found that fumaric acid” IDEM rephrase please.
“Occurrence and development of cancer cells” is another repetitive idea (12 times in 12 pages) that must be attended e.g. page 4: “promoting the occurrence and development of tumors [37-39]. Collectively, these studies confirmed that tumor cell metabolites can participate in the occurrence and development of tumors by regulating the signaling molecules in tumor cells”
Figure 1: Why is Gln employed in one section whereas in other “glutamine” ? same for arginine
Page 5: the acronym for pentose phosphate pathway was already stated and must be used here.
Figure 2 legend: remove a red _
NK acronym must be stated the first time of appearance (page 7).
Page 9: IDH acronym was previously stated.
Page 10 : “In addition, Treg cells in the TME can drive immunosuppression in the tumor microenvironment and mediate the immune” use the acronym and rephrase.
Page 13: the authors may want to explain why they mean by MCT destroyer…
Table 1: the authors must include the drug mechanism of action (e.g., inhibition of … activation of _target_) and separate and clearly state the effect in cancer cell and immune cell. Also, many of the proposed drugs are of questionable results in immune cells. E.g., Glutaminase inhibitor CB-839 enhances the activation of CD4 + Th1 and CD8 + CTL but suppress the differentiation of CD4+Th17 cells (PMID: 35897036) and inhibition of glycolysis by its inhibitor 2-DG (2-deoxyglucose) reduces cytotoxic capacity of T cells and the expression of key effector molecules such as IFNγ and granzymes, so this must be carefully and responsibly studied and mentioned for all the drugs in table 1.
Author Response
In this work Zhang and colleagues explore the metabolic reprogramming occurring during tumor development in the tumor microenvironment and its impact on the survival and function of immune cells. The topic is original considering that it includes 3 relevant topics of great interest nowadays: the impact of metabolic reprogramming in 1. the tumor microenvironment; 2. Cancer development and 3. immune cell survival and function. Despite being a very complex topic, the author's proposal contributes to this developing field.
In comparison to other reviews this manuscript integrates a more complex and physiological panorama, as most of the current reviews in these areas only explore a part (isolate) the topics e.g, the effect of tumors in the TME; or the interplay between the immune system and the tumor progression; the effect of metabolic reprogramming in cancer cells… but very few original works discuss the interaction between all these components. And I think that is a strength of the present manuscript.
However, despite the manuscript is adequate in the present form many aspects must be attended in the manuscript in order to improve the quality of this work and improve the audience appreciation.
Many observations mentioned by the authors as “nutrient deprivation”, “absence of amino acids in the cell culture” can promote an autophagic panorama. Alterations in autophagy (induction or inhibition) is another way to modulate cancer metabolism. Extensive and recent reviews are available regarding the role of autophagy in cancer cell metabolism, in the present manuscript the authors hardly (2 times) mention autophagy. The authors may want to extend and reinforce this idea.
Author's Response: Thanks to your suggestion, alteration (induction or inhibition) of autophagy is another way to regulate cancer metabolism. However, autophagy is not mentioned in this manuscript (2 times) to emphasize autophagy, but is briefly mentioned because cellular metabolic reprogramming mediates autophagy.
In abstract: “The occurrence and development of tumors require the metabolic reprogramming of cancer cells. Cancer cells can change their …” rephrase is encouraged, as it is repetitive in the present form.
Author's Response: Thank you for your assistance, it has been modified.
In abstract: “antitumor” is employed while in other parts of the text anti-tumor, please homogenize.
Author's Response: Thank you for your assistance, it has been modified.
In introduction: “Previous studies have found that the occurrence and development of tumors are closely related to the tumor microenvironment (TME). The abnormal accumulation of metabolites produced by the massive proliferation of tumor cells can promote the occurrence and development of tumors” rephrase to avoid repetitive text.
Author's Response: Thank you for your assistance, it has been modified.
Page 3: de novo, and other Latinism as in vivo, in vitro, etc., must be italicized
Author's Response: Thank you for your assistance, it has been modified.
Page 3: “convert lactic acid to pyruvate. Pyruvate can competitively bind to proline hydroxylase 2” IDEM, rephrase, perhaps as: to pyruvate. The latter can…
Author's Response: Thank you for your assistance, it has been modified.
Page 3: downstream-regulated gene 3 (NDRG3) and other genes must be italicized.
Author's Response: Thank you for your assistance, it has been modified.
Page 4: “Thus resulting in the accumulation of succinic acid in tumor cells. The accumulation of succinic acid” IDEM, rephrase please.
Author's Response: Thank you for your assistance, it has been modified.
Page 4: “action of fumaric acid found that fumaric acid” IDEM rephrase please.
Author's Response: Thank you for your assistance, it has been modified.
“Occurrence and development of cancer cells” is another repetitive idea (12 times in 12 pages) that must be attended e.g. page 4: “promoting the occurrence and development of tumors [37-39]. Collectively, these studies confirmed that tumor cell metabolites can participate in the occurrence and development of tumors by regulating the signaling molecules in tumor cells”
Author's Response: Thank you for your assistance, it has been modified.
Figure 1: Why is Gln employed in one section whereas in other “glutamine” ? same for arginine
Author's Response: Thank you for your assistance, it has been modified.
Page 5: the acronym for pentose phosphate pathway was already stated and must be used here.
Figure 2 legend: remove a red _.
Author's Response: Thank you for your assistance, it has been modified.
NK acronym must be stated the first time of appearance (page 7).
Author's Response: Thank you for your assistance, it has been modified.
Page 9: IDH acronym was previously stated.
Author's Response: Thank you for your assistance, it has been modified.
Page 10 : “In addition, Treg cells in the TME can drive immunosuppression in the tumor microenvironment and mediate the immune” use the acronym and rephrase.
Author's Response: Thank you for your assistance, it has been modified.
Page 13: the authors may want to explain why they mean by MCT destroy…
Author's Response: Thank you for your assistance, it has been modified. What the authors in the manuscript are trying to say is that this means that blocking MCT inhibits tumor cell proliferation while supporting immune cell activation
Table 1: the authors must include the drug mechanism of action (e.g., inhibition of … activation of _target_) and separate and clearly state the effect in cancer cell and immune cell. Also, many of the proposed drugs are of questionable results in immune cells. E.g., Glutaminase inhibitor CB-839 enhances the activation of CD4 + Th1 and CD8 + CTL but suppress the differentiation of CD4+Th17 cells (PMID: 35897036) and inhibition of glycolysis by its inhibitor 2-DG (2-deoxyglucose) reduces cytotoxic capacity of T cells and the expression of key effector molecules such as IFNγ and granzymes, so this must be carefully and responsibly studied and mentioned for all the drugs in table 1.
Author's Response: Thanks to your suggestion, authors provide the necessary additional information on each drug (in Chapter 6).